# Genome-Wide Transcriptional Profiling Reveals PHACTR1 as a Novel Molecular Target of Resveratrol in Endothelial Homeostasis

**DOI:** 10.3390/nu14214518

**Published:** 2022-10-27

**Authors:** Meiming Su, Wenqi Zhao, Yujie Li, Hong Li, Suowen Xu, Jianping Weng

**Affiliations:** 1Department of Endocrinology, Institute of Endocrine and Metabolic Diseases, The First Affiliated Hospital of USTC, Division of Life Sciences and Medicine, Clinical Research Hospital of Chinese Academy of Sciences (Hefei), University of Science and Technology of China, Hefei 230001, China; 2Department of Medical Biotechnology, School of Basic Medical Sciences, Guangzhou University of Chinese Medicine, Guangzhou 510006, China

**Keywords:** resveratrol, endothelial cell, inflammation, atherosclerosis, PHACTR1

## Abstract

Atherosclerosis is a chronic inflammatory vascular disease in which endothelial cells play an important role in maintaining vascular homeostasis. Endotheliitis caused by endothelial dysfunction (ED) is the key cause for the development of cardiovascular and cerebrovascular diseases as well as other vascular system diseases. Resveratrol (RES), a multi-functional polyphenol present in edible plants and fruits, prevents cardiovascular disease by regulating a variety of athero-relevant signaling pathways. By transcriptome profiling of RES-treated human umbilical vein endothelial cells (HUVECs) and in-depth bioinformatic analysis, we observed that differentially expressed genes (DEGs) were enriched in KEGG pathways of fluid shear stress and atherosclerosis, suggesting that the RES may serve as a good template for a shear stress mimetic drug that hold promise in combating atherosclerosis. A heat map and multiple datasets superimposed screening revealed that RES significantly down-regulated phosphatase and actin modulator 1 (PHACTR1), a pivotal coronary artery disease risk gene associated with endothelial inflammation and polyvascular diseases. We further demonstrate that RES down-regulated the gene and protein expression of PHACTR1 and inhibited TNF-α-induced adhesion of THP-1 monocytes to activated endothelial cells via suppressing the expression of PHACTR1. Taken together, our study reveals that PHACTR1 represents a new molecular target for RES to maintain endothelial cell homeostasis and prevent atherosclerotic cardiovascular disease.

## 1. Introduction

Atherosclerosis, a chronic inflammatory vascular disease, and the most common form of cardiovascular disease, remains one of the most morbid, mortal, and medically expensive diseases [1,2]. Atherosclerotic plaque formation is driven by several traditional and non-traditional risk factors, including high systolic blood pressure, high fasting plasma glucose, and high low-density lipoprotein cholesterol [3]. Atherosclerotic lesions are characterized by long-term accumulation and transformation of various cell types, lipids, and fragmented tissues [4]. Endothelial cells are “molecular bridges” connecting blood and vascular walls, thereby maintaining tissue homeostasis. Endotheliitis caused by endothelial dysfunction (ED) is the initiator and driver for the development of a plethora of cardiovascular and cerebrovascular diseases and other vascular system disorders [5]. Treatment drugs for ED include lipid-lowering drugs (such as statins), anti-hypertensive drugs (such as angiotensin converting enzyme inhibitor), anti-diabetic drugs (such as metformin), anti-inflammatory drugs (such as colchicine), nitric oxide synthase 3 (eNOS) enhancers, and many others [5]. Another category of endothelial protective drugs are shear stress mimetic drugs that mimic blood flow/shear stress related mechanosignaling pathways and prevent ED and atherosclerosis [6].

Resveratrol (3,4′,5-trihydroxystyrene, RES) is a polyphenolic compound found in various edible plants such as grapes and berries, and it possesses a wide variety of beneficial effects, including antioxidant, anti-inflammatory, and others [7,8,9]. Recent studies have shown that RES ameliorates ED and prevents cardiovascular disease via multiple mechanisms, which includes inhibition of oxidative stress and inflammatory responses, regulation of lipid metabolism, enhancement of insulin sensitivity, promotion of GLUT4 expression/translocation and ensuing glucose uptake, and activation of the SIRT1/AMPK signaling pathway and serving as a blood flow mimetic drug via activating Krüppel-like factor 2 (KLF2) [10,11,12,13]. However, the beneficial effects of RES in endotheliitis are still not well elucidated. Identification of new molecular targets of RES could deepen our understanding of the vasculoprotective effects of RES as well as provide new therapeutic targets of cardiovascular disease.

Genome-wide association studies (GWAS) have shown that phosphatase and actin regulator 1 (PHACTR1) are associated with a variety of vascular diseases, including atherosclerosis and carotid artery dissection [14,15]. In addition, PHACTR1 rs6905419 C allele may be related to muscle strength, evidenced by the positive association between PHACTR1 rs6905419 C allele and weightlifting performance in Russian athletes [16]. PHACTR1 is also involved in development, and Ito et al. found that PHACTR1 takes part in neuronal functions regulated in a spatiotemporal manner [17]. PHACTR1 is an actin and protein phosphatase 1 (PP1) binding protein that is reported to be highly expressed in the brain and controls PP1 activity and F-actin remodeling; the researchers found that PHACTR1 plays a central role in tube formation and the control of endothelial cell survival [18,19]. These studies reveal important physiological functions of PHACTR1. However, the role of PHACTR1 in cardiovascular disease is still controversial, especially its cell-type specificity. Western diet feeding increases mouse macrophage cytoplasmic hyperplasia defects and plaque necrosis in *Ldlr^−/−^* mice lacking hematopoietic PHACTR1 [20]. In vascular smooth muscle cells, loss of PHACTR1 does not appear to induce a pathological vascular phenotype, but overexpression of PHACTR1 promotes calcification of blood vessels [21,22]. However, studies have also shown that downregulation of PHACTR1 promotes expressions of the factors involved in atherosclerotic events, such as proinflammatory proteins and amino-sensitive amine oxidase [23]. At the same time, our recent study has revealed that PHACTR1 exerts a detrimental effect on endothelial function by inducing endothelial inflammation and reducing NO production, which provides mechanistic insights into the potential of RES to prevent ED and atherosclerosis by targeting PHACTR1 expression [24]. However, it remains unclear whether PHACTR1 expression can be modulated by naturally-occurring phytochemicals such as RES.

In this study, we extracted RNA from RES-treated HUVECs and performed genome-wide transcriptomic profiling, with an aim to identifying new targets of RES in protecting against cardiovascular diseases. The present study has important clinical implications for the prevention and treatment of cardiovascular diseases by addressing pleiotropic benefits exerted by RES.

## 2. Materials and Methods

### 2.1. Cell Culture

HUVECs isolated from the umbilical cords of healthy pregnant women are the standard model for studying endothelial cell function in vitro [25,26]. All subjects gave their informed consent for inclusion before they participated in the study. The study was conducted in accordance with the Declaration of Helsinki, and the protocol was approved by the Ethics Committee of the institutional review board of USTC (2020-ky013). In this study, 3rd- to 7th-generation HUVECs from different donors were used. The cells were cultured in ECM endothelial cell growth medium (ScienCell, Carlsbad, CA, USA) containing 1× endothelial cell growth supplement (ScienCell, Carlsbad, CA, USA), 5% fetal bovine serum, and 1% penicillin/streptomycin antibiotics. The culture conditions were 37 °C and 5% CO_2_. HUVECs seeded in 12-well plates were treated with DMSO or 1 μM, 2.5 μM, 5 μM, 10 μM, 20 μM, and 25 μM of RES (#501-36-0, TargetMol, Shanghai, China) for 12 h or 24 h after cells reached confluence. For experiments in which RES affected PHACTR1 expression and functional experiments, HUVECs were also infected with Ad-PHACTR1 (MOI = 1, #NM_030948, WZ Biosciences, Shandong, China) or control adenovirus for 24 h, with/without 10 ng/mL of TNF-α (#300-01A, PeproTech, Rocky Hill, NJ, USA) treatment for 6 h.

### 2.2. Total RNA Extraction, RNA-Sequencing, and Quantitative Real-Time PCR

An amount of 20 μM of RES or DMSO was used to treat HUVECs for 24 h. After treatment, RNA was isolated using RNeasy kits (Yishan Biotech, Shanghai, China). After quality control of RNA, a library was constructed at the Beijing Genomics Institute (BGI) in Shenzhen, China for RNA sequencing (RNA-seq). Briefly, mRNA molecules were isolated from total RNA using oligo(dT)-attached magnetic beads. mRNA molecules were fragmented into small pieces using fragmentation reagent after reaction for a certain period at the proper temperature. First-strand cDNA was generated using random hexamer-primed reverse transcription, followed by a second-strand cDNA synthesis. The synthesized cDNA was subjected to end-repair and was then 3′ adenylated. Adapters were ligated to the ends of these 3′ adenylated cDNA fragments. The cDNA fragment was amplified by PCR, and the products were purified with Ampure XP Beads (AGENCOURT, Thermo Fisher Scientific, Waltham, MA, USA), and dissolved in EB solution. The library was validated on the Agilent Technologies 2100 bioanalyzer (Santa Clara, CA, USA). The double stranded PCR products were heat denatured and circularized by the splint oligo sequence. The single strand circle DNA (ssCir DNA) were formatted as the final library. The library was amplified with phi29 to make a DNA nanoball (DNB), which had more than 300 copies of one molecular. The DNBs were loaded into the patterned nanoarray, and single-end 50 (pair end 100/150) base reads were generated by way of combinatorial Probe-Anchor Synthesis (cPAS). The mRNA expression profiles of GSE211883 (human) were obtained from the Gene Expression Omnibus (GEO) database (https://www.ncbi.nlm.nih.gov/geo/query/acc.cgi?acc=GSE211883) accessed on 26 August 2022. The samples of the DMSO-treated control group and the RES-treated group were in triplicate. Reverse transcription kits (TaKaRa, Dalian, China) were used to reverse transcribe total RNA into complementary DNA. After the reverse transcription process, real-time PCR detection was performed in the Roche LC96 real-time PCR detection system using TB Green^®^ Premix Ex Taq™ II (Tli RNaseH Plus) (TaKaRa, Dalian, China). The relative expression of RNA was calculated by the 2^(−ΔΔCt)^ method, and GAPDH was used as a normalization control. The sequences of the primers are shown in Appendix A.

### 2.3. Test for Correlation and Variation of Samples

Pearson correlation coefficient analysis and principal component analysis (PCA) were performed on the mRNA expression profile of the GSE211883 dataset using OmicShare Tools to test the correlation and variability among samples. Pearson correlation coefficients are used to measure linear relationships between samples, using the pheamap software package to plot hierarchical cluster heatmaps for visualization of the correlation of gene expression across all pairs of sample combinations in a dataset. PCA is used to check how similar replicated samples are to each other (clustering) and ensure that experimental conditions are the primary source of data variation.

### 2.4. Identification of DEGs

BGI used the limma software package in R to normalize and screen DEGs between control and RES samples. The *q*-value < 0.05 for DEGs were filtered as significant. Heatmaps were drawn using the pheatmap software package of the OmicShare Tools Bioinformatics Cloud Platform (https://www.omicshare.com/tools) for visualization of DEGs.

### 2.5. Pathway Enrichment Analysis (KEGG) and Gene Ontology (GO) Analysis

Gene Ontology (GO) [27] and Kyoto Encyclopedia of Genes and Genomes (KEGG) [28] pathway enrichment analysis were performed using the clusterProfiler software package in OmicShare Tools. The filter condition is *q*-values < 0.05.

### 2.6. DEGs Analysis in Discovery Cohorts

Microarray datasets were collected from the GEO (https://www.ncbi.nlm.nih.gov/gds/) database and analyzed using the online software GEO2R (https://www.ncbi.nlm.nih.gov/geo/geo2r/). The GEO dataset used in this study was GSE41571, which contained 6 stable plaque samples and 5 ruptured plaque samples [29]. The differences in PHACTR1 expression in these 11 samples were analyzed.

### 2.7. Protein Extraction and Western Blotting Analysis

HUVECs were treated with vehicle (DMSO) or RES (1 μM, 2.5 μM, 5 μM and 10 μM) for 12 h, followed by adenovirus (MOI = 1) infection for another 24 h to overexpress PHACTR1. The treated cells were washed three times with ice-cold PBS buffer to remove the cell culture medium. Whole cell lysate was extracted from HUVECs with 1× sample buffer and boiled at 95 °C for 10 min. Samples were separated by SDS-PAGE and then transferred to nitrocellulose membranes (Pall, New York, NY, USA) and then incubated with blocking buffer (Li-COR, Lincoln, NE, USA) for 1 h at room temperature. After blocking, primary antibody was added and incubated overnight at 4 °C. The primary antibodies used are listed in Appendix A. The cells were washed 3 times using 1× Tris buffer saline (TBS) containing 0.1% Tween-20 for 10 min each. The membrane was then incubated for 1 h at room temperature with IRDye 680RD goat anti-murine IgG (H+L) or IRDye 800CW goat anti-rabbit IgG (H+L) (1:10,000 dilution, Li-COR, Lincoln, NE, USA). Finally, the blots were visualized using the Li-COR CLx infrared imaging system.

### 2.8. Statistical Analysis

This study used GraphPad Prism software version 8.0 (GraphPad software, La Jolla, CA, USA) for plotting and statistical analysis. Data normality was determined by the Shapiro–Wilk test. Comparisons between groups were performed using unpaired 2-tailed t tests or one way ANOVA with Bonferroni corrections for data that passed the normality test. Data are expressed as means ± SD. A *p*-value < 0.05 was considered statistically significant.

## 3. Results

### 3.1. Dataset Validation and Identification of DEGs

As shown in the flowchart in Figure 1A, RNA was extracted from RES- and DMSO-treated HUVECs, and transcriptome sequencing (GSE211883) was performed. By using bio-informatics research methods such as the validation of data correlation, the screening of DEGs, and functional enrichment analysis, DEGs and signaling pathways between the two groups were studied. At the same time, DEGs were overlaid using the GWAS-CAD database and the UK Biobank database and the screened genes were validated with the GSE41571 dataset [30,31,32]. Finally, the expression of screened genes was verified by real-time PCR and Western blotting, and the role of the target gene was determined using cell adhesion assay.

Pearson-related tests and PCA were used to validate the dataset. The relevant heatmap of the GSE211883 dataset shows a strong correlation between samples in each group (Figure 1B). The box pattern shows that the mRNA expression distribution in samples is consistent (Figure 1C). PCA of GSE211883 showed that 6 samples in the two groups were separated well, the control group’s samples were very close in the PC1 and PC2 dimensions, and the RES group was also very close to each other (Figure 1D). A *q*-value < 0.05 was set, and a total of 1319 DEGs were identified. Compared with the control group, a total of 819 upregulated and 500 downregulated DEGs were found in the RES group, as shown in the DEGs statistics (Figure 1E), the volcano diagram (Figure 1F), and the heatmap (Figure 1G). The top 10 DEGs are listed in Table 1. Solute carrier family 52 member 1 (*SLC52A1)* and cytochrome p450 family 1 subfamily A member 1 (*CYP1A1)* were the most significantly up- and downregulated genes in the RES samples, respectively.

### 3.2. Functional and Pathway Enrichment Analysis

KEGG pathway enrichment analysis showed that DEGs were mainly enriched in the p53 signaling pathway, apoptosis, MAPK signaling pathway, arginine and proline metabolism, fluid shear stress, and atherosclerosis (Figure 2A,B). The top 10 pathways of the DEGs enrichment analysis by *q*-value are listed in Table 2. There are many key molecules involved in the laminar shear stress and atherosclerosis pathways, such as endothelin 1 (*EDN1*), vascular cell adhesion molecule 1 (*VCAM1*), *KLF2*, intercellular adhesion molecule 1 (*ICAM1)*, and platelet and endothelial cell adhesion molecule 1 (*PECAM1)*. Furthermore, the GO analysis results (Figure 2C,D) showed that DEGs associated with biological processes were significantly enriched in cellular processes, bioregulation, and metabolic processes (Figure 2C). As for the cell composition, DEGs were significantly enriched in cells, organelles, and cell membranes. Additionally, the molecular function of DEGs were found to be enriched in the binding, catalytic activity, and regulation of molecular function.

### 3.3. Multi-Dataset Screening Showed That PHACTR1 Might Be Associated with the Vascular Protective Effects of RES

In total, 1319 DEGs were overlapped with the other two published coronary artery disease (CAD)-related datasets from GWAS-CAD [31] and UK Biobank [32] (Figure 3A). A total of seven genes were found, namely, ldl receptor related protein 1 (*LRP1)*, coiled-coil domain containing 92 (*CCDC92)*, tribbles pseudokinase 1 (*TRIB1)*, transforming growth factor beta 1 (*TGFB1)*, *PECAM1*, serpin family H member 1 (*SERPINH1)*, and *PHACTR1* (Table 3). Among them, *LRP1* and *PHACTR1* show the most significant difference between the two groups in sequencing results, and it has been reported that RES can upregulate LRP1 protein expression in mouse brains [33]. Subsequently, heatmap analysis was performed on 1319 DEGs, and it was found that after RES treatment of endothelial cells, *KLF2* was significantly upregulated and *EDN1* and *PHACTR1* were significantly downregulated (Figure 3B). HUVECs were treated with DMSO or various concentrations of RES (10 μM and 20 μM), and the sequencing results were validated by real-time PCR. It was found that RES was able to upregulate the mRNA level of *KLF2* and downregulate the mRNA level of *EDN1* in a concentration-dependent manner (Figure 3C,D). In endothelial cells, the mechano-responsive transcription factor KLF2 is an important regulator of intravascular homeostasis and can participate in the prevention of atherosclerosis through anti-inflammatory action [34]. This is consistent with the result of a recently published article showing that RES acts as an agonist of KLF2 in endothelial cells, exerting a protective effect [35]. Upregulation of *EDN1* (ET1) is critically involved in inflammation/vasoconstriction, and the very early stage of plaque evolution, whereas increased secretion of ET1 has also been reported to accelerate the pathogenesis of atherosclerosis [36,37]. RES significantly attenuated ET1-induced protein tyrosine phosphorylation in smooth muscle cells [38]. In endothelial cells, RES interferes with the ERK1/2 pathway by blocking reactive oxygen species formation and ultimately inhibits strain-induced ET1 expression [39]. Because the effects of RES on KLF2 and EDN1 have been reported previously, we focused on the effects of RES on PHACTR1 expression. We speculate that PHACTR1 may be a new molecular target of RES in maintaining endothelial cell homeostasis and preventing cardiovascular disease.

Through the mining and analysis of the published dataset in the NCBI GEO database, we found that the expression of PHACTR1 in the ruptured carotid artery showed an upwards trend in the GSE41571 dataset compared with the stable carotid artery, suggesting that PHACTR1 may be related to the occurrence and development of atherosclerosis. This analysis also further demonstrates the correlation between PHACTR1 and cardiovascular disease (Figure 3E,F). The effect that RES downregulated the mRNA level of PHACTR1 was further verified by real-time PCR (Figure 3G).

### 3.4. PHACTR1 Expression Was Downregulated in RES-Mediated Anti-Inflammatory Effects

Recently, we reported that overexpression of PHACTR1 can promote endothelial inflammation and monocyte adhesion [24]. Previous studies have also shown that RES can inhibit monocyte adhesion to endothelial cells [51]. Therefore, we investigated the effect of RES on PHACTR1 expression and whether down-regulation of PHACTR1 mediated the effect of RES on monocyte adhesion to endothelial cells. As shown in Figure 4A, TNF-α promotes the expression of *PHACTR1*, and the effect could be reversed by RES. We then treated endothelial cells with different doses of RES for 12 h, followed by adenovirus infection for another 24 h to overexpress PHACTR1. The results showed that RES was able to downregulate the protein expression of PHACTR1 in a concentration-dependent manner (Figure 4B,C). To further explore the role of PHACTR1 in the anti-inflammatory function of RES in endothelial cells, we conducted THP-1 cell adhesion experiments. Adhesion of THP-1 monocytes to endothelial cells is also an important mechanism that drives endothelial cell inflammation [52]. We observed that the use of TNF-α alone significantly stimulated THP-1 cell adhesion to endothelial cells, which was evidently reduced by RES. Intriguingly, the inhibitory effect of RES was blocked by overexpression of PHACTR1 (Figure 4D,E). These results suggest that RES may exert an anti-inflammatory effect on endothelial cells by reducing PHACTR1 gene and protein expression.

## 4. Discussion

PHACTR1 is a GWAS-identified gene associated with polyvascular diseases, including coronary artery disease and dissection [31]. However, thus far, no therapeutic drugs have been available that can regulate PHACTR1 expression. Our study demonstrates for the first time that the CVD risk-related molecule PHACTR1 is regulated by natural products RES. In addition, we report that RES may protect endothelial cell dysfunction (evidenced by reduced monocyte adhesion to endothelial cells) by reducing PHACTR1 expression.

As a natural product present in grapes, berries, and other plant sources, RES plays a protective role such as antioxidant and anti-inflammatory in many diseases, especially cardiovascular disease [53,54]. In previous studies, RES has been found to increase the expression and activity of eNOS in endothelial cells and activate SIRT1 as well as the expression of KLF2 [35,55,56]. Luo et al. elegantly showed that RES inhibited endothelial inflammation and atherosclerosis in mice by upregulating the expression of uncoupling protein 2 through transcription factor KLF2 [57]. The protective mechanism of RES on cardiovascular disease has always been the focus and hotspot of research. By performing transcriptome sequencing and bioinformatics analysis on RES-treated endothelial cells, new molecular targets can be identified, which may further explain the cardioprotective effects of RES.

We found that SLC52A1 and CYP1A1 were the most significantly up- and downregulated gene in RES-treated human endothelial cells, respectively. This was consistent with the RNA sequencing results observed in fibroblasts, which showed that SLC52A1 was significantly up-regulated after RES treatment [58]. Studies have shown that CYP1A1 may be involved in the pathogenesis of atherosclerosis and the occurrence of diabetes mellitus and its vascular complications [59]. RES significantly downregulated the expression of CYP1A1 in endothelial cells and HepG2 cells, in which the activity of the CYP1A1 promoter was also inhibited [60].

In the analysis of biological function enrichment of DEGs, we found that RES was closely related to the p53 signaling pathway and apoptosis, and RES significantly upregulated the expression of tumor protein p53, Fas cell surface death receptor, BCL2 associated X, Fos proto-oncogene, Jun proto-oncogene, and other genes. Hsieh et al. found that RES was able to increase eNOS expression/activity, induce the accumulation of p53 and p21, and thus inhibit the proliferation of bovine pulmonary artery endothelial cells [61]. She et al. has also demonstrated that RES can induce activation of p53 as well as apoptosis [62]. These data suggest that RES has a wide range of anticancer effects, but on the contrary, accumulating studies have also showed that RES has a profound anti-apoptosis effect. Zhang et al. demonstrated that RES increases the expression of SIRT1 and boosts antioxidant capacity, thereby reducing mitochondrial-associated apoptosis signaling pathways and preventing ROS-induced myoblast damage [63]. The controversial role of RES in the proliferation and apoptosis of endothelial cells may depend on the dose of RES. Zhang et al. conducted cytotoxicity experiments on HUVECs treated with RES and found that at a concentration of 300 μM and higher, RES was cytotoxic, as the cell viability was significantly inhibited [64].

In the KEGG signaling pathway analysis of DEGs, we found that RES was associated with fluid shear stress and atherosclerosis, including VCAM1, ICAM1, PECAM1, and other adhesion molecules closely related to atherosclerosis. Mattison et al. found that in rhesus monkeys RES prevented arterial wall inflammation induced by a high-fat and high-sucrose diet and reduced the aortic pulse wave velocity [65]. RES administration in mice by Kaneko et al. prevents the development of CaCl_2_-induced abdominal aortic aneurysms and reduces the expression of inflammatory-related factors such as TNF-α, CD68, and ICAM1 [66]. Deng et al. found that RES can inhibit the adhesion of monocytes to endothelial cells induced by TNF-α and reduce the expression of adhesion molecules such as VCAM1 and ICAM1 [67]. However, unlike previous RES anti-inflammatory studies, our sequencing results indicate that RES significantly upregulates the expression of inflammatory-related factors such as VCAM1, ICAM1, and PECAM1. Differently, we found that RES has a relieving effect on the TNF-α-induced adhesion of THP-1 cells and endothelial cells, which is consistent with the known anti-inflammatory effects of RES. Thus, we speculate that, in the absence of an exogenous inflammation stimulus, the mild upregulation of inflammation associated genes may be compensatory.

GWAS have shown that PHACTR1 is associated with a variety of vascular diseases, including atherosclerosis and carotid artery dissection [14,15]. However, the role of PHACTR1 in endothelial function and cardiovascular disease is still unclear. PHACTR1 may exacerbate endothelial cell oxidative stress and the inflammatory response through NF-κB signaling pathway [68]. However, other studies have shown that downregulating PHACTR1 promotes the factors involved in atherosclerotic events, such as proinflammatory proteins [23]. Through mining datasets in the GEO database, we found that the expression of PHACTR1 has a trend of being increased in the ruptured carotid artery, indicating its role in the occurrence and development of atherosclerosis. This piece of data agrees with another GEO dataset of human stable and vulnerable/ruptured plaque tissues, as we have reported recently [24]. In the recent study, we demonstrated that overexpression of PHACTR1 promoted endothelial inflammation and monocyte adhesion [24]. Previous studies have also shown that RES can inhibit the adhesion of monocytes to endothelial cells [51]. Therefore, we investigated the effect of RES on PHACTR1 expression and whether down-regulation of PHACTR1 mediated the effect of RES on monocyte adhesion to endothelial cells. Our results show that RES significantly downregulates the gene and protein expression of PHACTR1 in a concentration-dependent manner. According to the analysis of functional experimental results, RES alleviated the adhesion of TNF-α-treated THP-1 monocytes to endothelial cells, while overexpression of PHACTR1 reversed this inhibitory effect and showed a significant role in promoting the adhesion of THP-1 monocytes to endothelial cells. These results suggest that RES may protect endothelial cell dysfunction by reducing PHACTR1 expression to prevent monocyte adhesion. These results provide valuable clues for further study of the pharmacological profile of RES in the treatment of cardiovascular disease. Further studies are needed to validate whether PHACTR1 expression is decreased in plaque tissues from patients with cardiovascular diseases. Previous clinical studies have shown that the use of RES can increase the expression of eNOS, promote the production of NO, and reduce the expression of ET1, indicating that RES has a good effect on FMD [69,70] (https://clinicaltrials.gov/, NCT04449198, NCT03436992, NCT03762096). At the same time, RES can also reduce the expression of inflammatory indicators in serum, indicating that RES has a good anti-inflammatory effect [71,72,73] (https://clinicaltrials.gov/, NCT01038089, NCT02244879). Based on imaging and epidemiologic evidence, RES has also been shown to improve vascular stiffness [74]. It has been shown that PHACTR1 can regulate the expression of ET-1 as well as the expression of inflammatory factors [24,75]. Our study suggests a new mechanistic explanation for the anti-inflammatory effects of RES.

We recognize that there are still some limitations in the study that should be noted. First, we did not explore whether PHACTR1 is a novel molecular target of RES in vivo, such as in *ApoE*^−/−^ mice fed with a Western-type diet. Second, the sample size used for analytical validation was small, which may affect the accuracy of the analysis results. Finally, future studies need to further expand the sample size and further investigate the specific mechanisms of PHACTR1 in cardiovascular disease through in vivo and in vitro experiments.

## 5. Conclusions

Our findings suggest that RES exerts a protective effect on endothelial cell dysfunction by preventing the adhesion of monocytes to endothelial cells via reducing the expression of PHACTR1, which may lead to the development of a new adjuvant therapy for atherosclerotic cardiovascular diseases.

## Figures and Tables

**Figure 1 nutrients-14-04518-f001:**
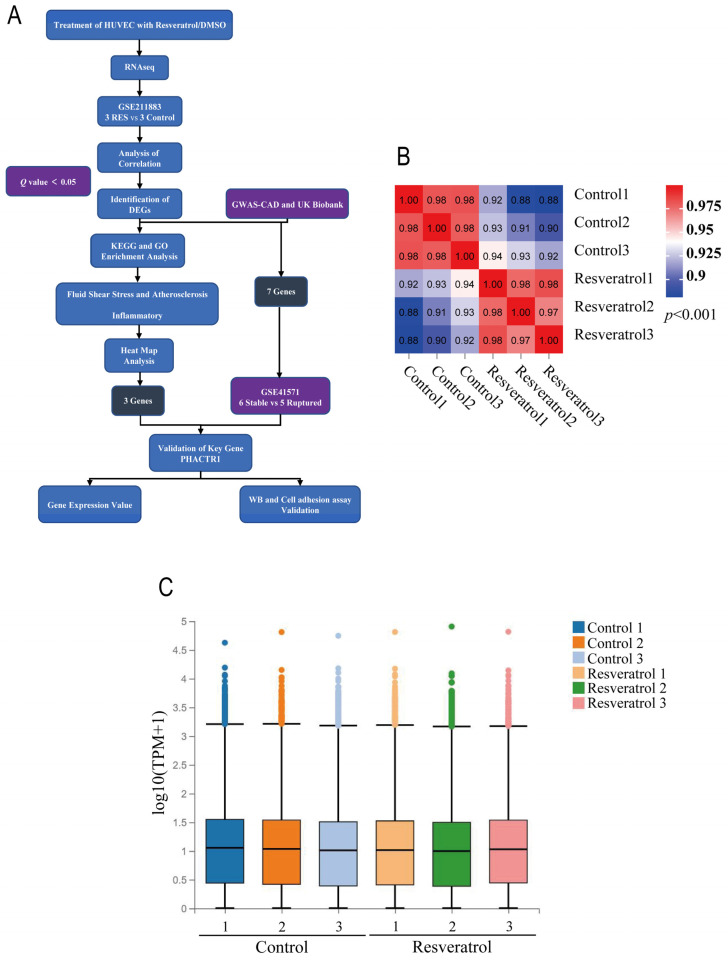
Dataset Validation and Identification of DEGs. (**A**) The flowchart of the study. **(B**) Pearson’s correlation analysis of samples from the GSE211883 dataset. The correlation coefficient is reflected by the colors in the heatmap. A correlation coefficient closer to 1 indicates a higher similarity between samples. (**C**) Box diagram of expression distribution. (**D**) PCA of samples from the GSE211883 dataset. PC1 and PC2 are represented on the x-axis and y-axis, respectively. PCA, principal component analysis; PC1, principal component 1; PC2, principal component 2. (**E**) The number of up- and downregulated DEGs (*p*-value < 0.001). (**F**) Volcano plot showing the DEGs between control and RES groups after analysis of the GSE211883 dataset with R software. The x-axis represents the fold-change (log-scaled) and the y-axis represents the *q*-value (log-scaled). Yellow symbols represent upregulated genes, red symbols represent downregulated genes. (**G**) A heatmap showing the DEGs between the two groups. Upregulated genes are labeled in red and downregulated genes are shown in blue.

**Figure 2 nutrients-14-04518-f002:**
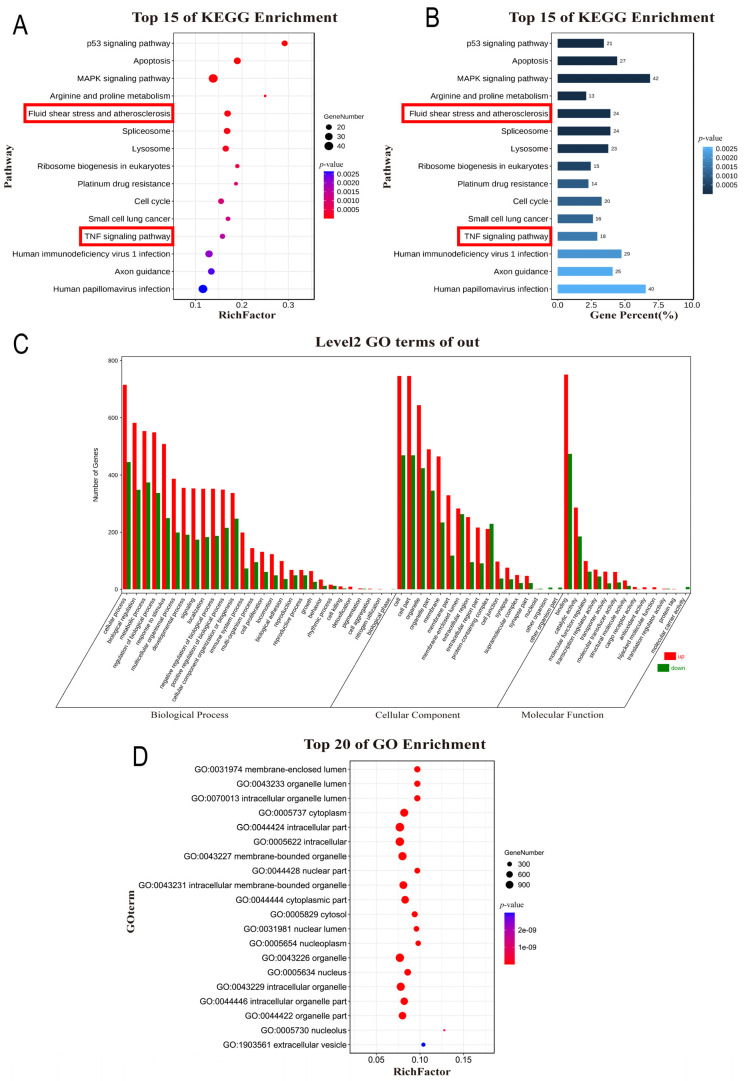
KEGG enrichment analysis and GO analysis. (**A**) Bubble diagram of the KEGG pathway enrichment analysis of DEGs. (**B**) Bar graph of the KEGG pathway enrichment analysis of DEGs. (**C**,**D**) GO functional enrichment analysis of DEGs. DEGs, Differentially Expressed Genes; BP, Biological Process; CC, Cellular Component; MF, Molecular Function.

**Figure 3 nutrients-14-04518-f003:**
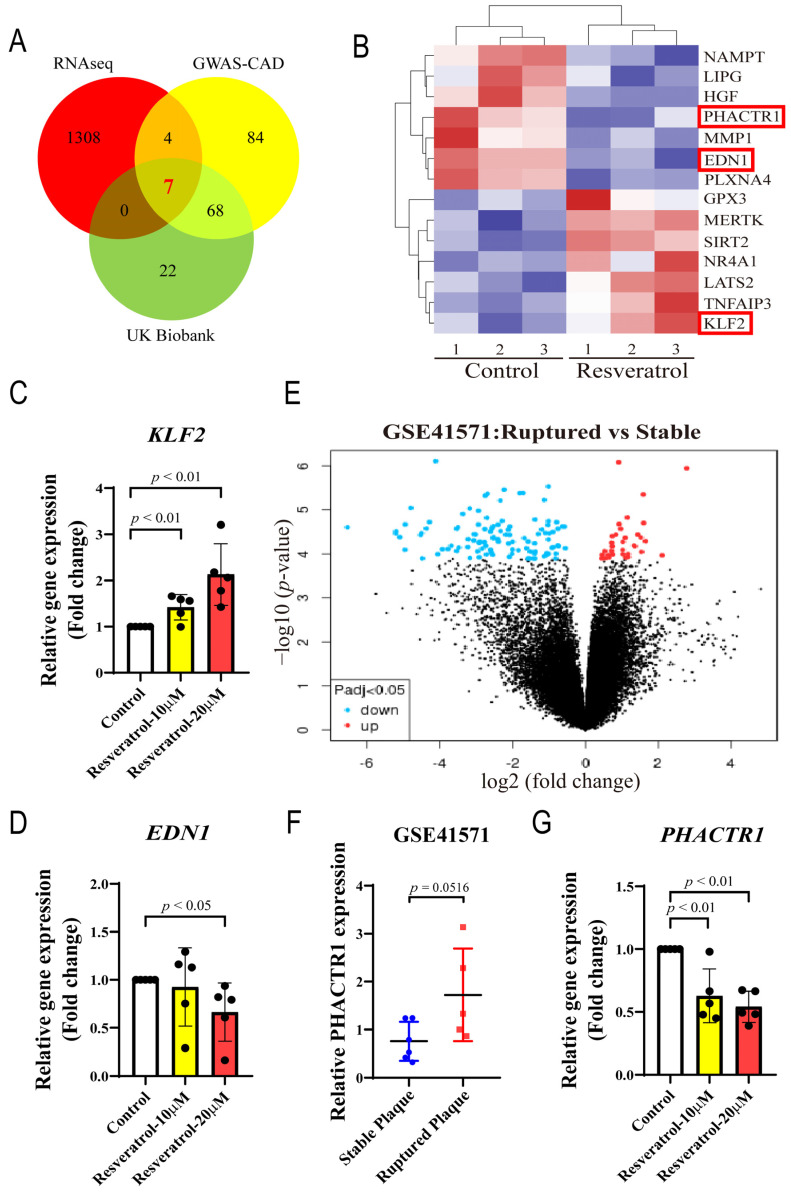
Systems-biology analysis of RES-treated human endothelial cells implicates PHACTR1 downregulation as the potential atheroprotective mechanism. (**A**) Venn diagram of DEGs overlapped with 2 datasets from GWAS-CAD [31] and UK Biobank [32]. (**B**) A heatmap showing the DEGs between the two groups. Upregulated genes are labeled in red and downregulated genes are shown in blue. (**C**) HUVECs were incubated with vehicle (DMSO) or RES (10 μM and 20 μM) for 24 h; qRT-PCR was then used to evaluate the mRNA level of *KLF2* in HUVECs (*n* = 5 biological replicates). (**D**) HUVECs were incubated with vehicle (DMSO) or RES (10 μM and 20 μM) for 24 h, qRT-PCR was then used to evaluate the mRNA level of *EDN1* in HUVECs (*n* = 5 biological replicates). (**E**) The volcano plot of DEGs from GSE41571 dataset. (**F**) Expression of *PHACTR1* mRNA levels in ruptured plaques (*n* = 5) and stable plaques (*n* = 6) from human plaques mined from GSE41571 dataset [30]. (**G**) HUVECs were incubated with vehicle (DMSO) or RES (10 μM and 20 μM) for 24 h; qRT-PCR was then used to evaluate the mRNA level of *PHACTR1* in HUVECs (*n* = 5 biological replicates). Summary plot depict the means ± SD.

**Figure 4 nutrients-14-04518-f004:**
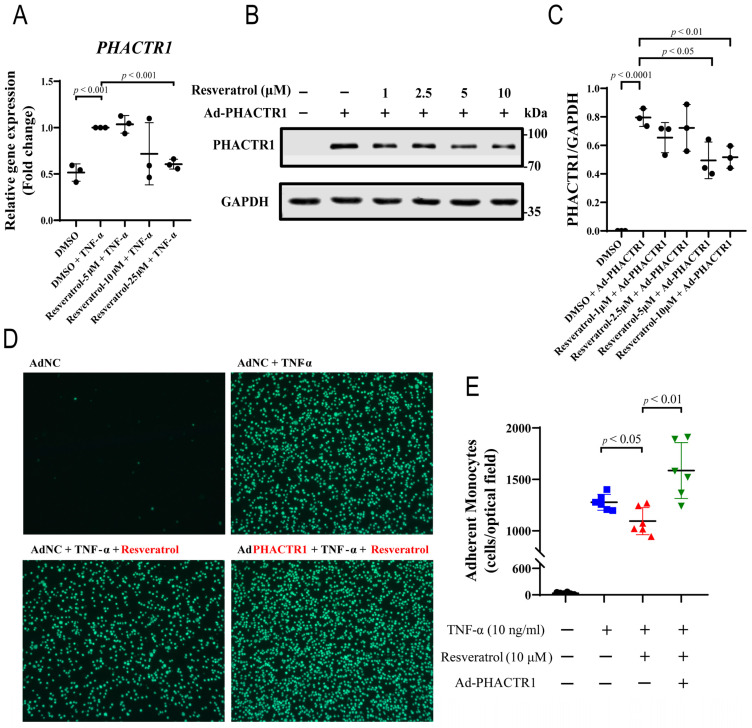
PHACTR1 Reverses RES-mediated Anti-inflammatory Effects. (**A**) HUVECs were pre-incubated with vehicle (DMSO) or RES (5 μM, 10 μM, and 25 μM) for 18 h and incubated with TNF-α (10 ng/mL) for 6 h; qRT-PCR was then used to evaluate the mRNA level of *PHACTR1* in HUVECs induced by TNF-α (*n* = 3 biological replicates). (**B**,**C**) HUVECs were pre-incubated with vehicle (DMSO) or RES (1 μM, 2.5 μM, 5 μM, and 10 μM) for 12 h and incubated with adenovirus infection to overexpress PHACTR1 for 24 h. Western blots were then conducted to evaluate the expression level of PHACTR1 (*n* = 3 biological replicates). For A–C, GAPDH was used as the internal reference. (**D**,**E**) PHACTR1 reversed the inhibition effect of RES on the adhesion of monocytes and endothelial cells. HUVECs were pre-treated with DMSO (black) or RES (10 μM) (red) for 12 h, further treated with control adenovirus or Ad-PHACTR1 (green) for 24 h as indicated, and incubated with TNF-α (10 ng/mL) (blue) for 6 h before THP-1 monocyte adhesion assay (*n* = 6 biological replicates). Original magnification ×10. Summary plots depict the means ± SD.

**Table 1 nutrients-14-04518-t001:** List of most significantly up- and downregulated genes by RES.

Gene Name	Gene Symbol	*q*-Value	Log2 (FC)
DEGs (upregulated)			
Solute Carrier Family 52 Member 1	*SLC52A1*	9.02 × 10^−9^	8.310
Epsin 3	*EPN3*	7.20 × 10^−7^	7.721
Grainyhead-like Transcription Factor 3	*GRHL3*	2.85 × 10^−6^	7.538
Nectin Cell Adhesion Molecule 4	*NECTIN4*	3.10 × 10^−5^	7.022
Ras Related Glycolysis Inhibitor and Calcium Channel Regulator	*RRAD*	1.66 × 10^−27^	6.789
Triggering Receptor Expressed on Myeloid Cell-like 1	*TREML1*	5.30 × 10^−4^	6.495
Transmembrane Protein 40	*TMEM40*	1.45 × 10^−3^	6.382
Lipocalin 15	*LCN15*	3.04 × 10^−3^	6.157
Dorsal Inhibitory Axon Guidance Protein	*DRAXIN*	2.17 × 10^−49^	6.147
Glutaminase 2	*GLS2*	2.23 × 10^−3^	5.824
DEGs (downregulated)			
Cytochrome P450 Family 1 Subfamily A Member 1	*CYP1A1*	1.46 × 10^−25^	−6.700
Hepatocyte Growth Factor	*HGF*	1.47 × 10^−2^	−4.303
Gap Junction Protein Alpha 5	*GJA5*	8.07 × 10^−3^	−2.885
Shisa Family Member 2	*SHISA2*	9.72 × 10^−3^	−2.368
5-Hydroxytryptamine Receptor 2B	*HTR2B*	1.40 × 10^−2^	−2.225
Dynein Axonemal Heavy Chain 8	*DNAH8*	3.45 × 10^−2^	−2.155
Vasoactive Intestinal Peptide Receptor 1	*VIPR1*	5.58 × 10^−4^	−1.879
Tolloid-like 1	*TLL1*	5.68 × 10^−4^	−1.819
Matrix Metallopeptidase 1	*MMP1*	3.04 × 10^−2^	−1.814
4-Hydroxyphenylpyruvate Dioxygenase-like	*HPDL*	3.17 × 10^−7^	−1.640

DEGs, Differentially expressed genes; FC, Fold change; RES, resveratrol.

**Table 2 nutrients-14-04518-t002:** The top 10 results of KEGG pathway enrichment analysis for DEGs.

Term	Count	Gene	*q*-Value
p53 signaling pathway	21	*PMAIP1*, *FAS*, *THBS1*, *BCL2L1*, *SESN2*, *SESN1*, *TP53I3*, *CCNB1*, *GADD45A*, *CD82*, *PPM1D*, *CDKN1A*, *DDB2*, *BBC3*, *BAX*, *AIFM2*, *TP53*, *CCND3*, *CDK2*, *PIDD1*, *RRM2*	7.65 × 10^−6^
Apoptosis	27	*CSF2RB*, *PMAIP1*, *FOS*, *FAS*, *ITPR3*, *BCL2L1*, *CTSZ*, *BIRC3*, *GADD45A*, *TRAF1*, *ITPR2*, *BBC3*, *BAX*, *CTSV*, *NFKBIA*, *LMNB2*, *RIPK1*, *TP53*, *CTSC*, *CTSD*, *PARP3*, *DFFB*, *DDIT3*, *PIDD1*, *JUN*, *AKT3*, *NRAS*	5.82 × 10^−4^
MAPK signaling pathway	42	*PDGFA*, *RAC2*, *FOS*, *FAS*, *NF1*, *RPS6KA1*, *HGF*, *EPHA2*, *ARRB1*, *PTPRR*, *CACNA2D1*, *PGF*, *NFKB2*, *HSPA8*, *PLA2G4C*, *GADD45A*, *KITLG*, *MAP3K4*, *RELB*, *DUSP1*, *DUSP5*, *RASA1*, *MAPK8IP1*, *JUND*, *CSF1*, *RRAS*, *FGFR3*, *IL1A*, *FLNC*, *TP53*, *TGFB1*, *PLA2G4A*, *ANGPT2*, *NR4A1*, *PRKACB*, *ARRB2*, *TAOK1*, *DDIT3*, *JUN*, *AKT3*, *NRAS*, *PTPN7*	4.86 × 10^−3^
Arginine and proline metabolism	13	*SMOX*, *ALDH4A1*, *CKB*, *SMS*, *ALDH7A1*, *SAT1*, *AMD1*, *GAMT*, *ALDH18A1*, *CARNS1*, *P4HA2*, *PYCR1*, *SRM*	5.54 × 10^−3^
**Fluid shear stress and atherosclerosis**	24	*PDGFA*, *RAC2*, *EDN1*, *FOS*, *SDC4*, *GSTT2B*, *HSP90AB1*, *CYBA*, *GSTM2*, *VCAM1*, *PTK2*, *DUSP1*, *KLF2*, *SQSTM1*, *ICAM1*, *TRPV4*, *GPC1*, *IL1A*, *TP53*, *PECAM1*, *RHOC*, *SDC1*, *JUN*, *AKT3*	5.54 × 10^−3^
Spliceosome	24	*GGT1*, *TRA2A*, *PPIL1*, *SNRPB*, *DHX15*, *PRPF19*, *SRSF10*, *SRSF3*, *HSPA8*, *HNRNPK*, *SRSF7*, *TRA2B*, *HNRNPC*, *DDX39B*, *HNRNPM*, *SNRPA1*, *SRSF2*, *SRSF1*, *HNRNPA1*, *HNRNPU*, *TCERG1*, *DDX5*, *PRPF40A*, *HNRNPA3*	5.54 × 10^−3^
Lysosome	23	*NPC1*, *HYAL1*, *GUSB*, *NAGA*, *LAPTM5*, *LGMN*, *CTSZ*, *NPC2*, *ARSA*, *ABCA2*, *LAMP3*, *CTSV*, *PSAP*, *NEU1*, *AP3B1*, *CD68*, *MAN2B1*, *CTSC*, *CTSD*, *ABCB9*, *GBA*, *TPP1*, *FUCA1*	8.19 × 10^−3^
Ribosome biogenesis in eukaryotes	15	*XRN2*, *NOP56*, *BMS1*, *NAT10*, *GNL3*, *DKC1*, *IMP4*, *RAN*, *WDR3*, *NHP2*, *LSG1*, *WDR43*, *GAR1*, *HEATR1*, *XPO1*	2.13 × 10^−2^
Platinum drug resistance	14	*PMAIP1*, *FAS*, *BCL2L1*, *GSTT2B*, *BIRC3*, *BRCA1*, *GSTM2*, *CDKN1A*, *BBC3*, *BAX*, *POLH*, *TP53*, *TOP2B*, *AKT3*	3.43 × 10^−2^
Cell cycle	20	*ORC6*, *CDC6*, *PCNA*, *CDC27*, *PKMYT1*, *CCNB1*, *STAG1*, *GADD45A*, *CDKN1A*, *MCM6*, *PLK1*, *E2F4*, *TP53*, *TGFB1*, *DBF4*, *CUL1*, *CCND3*, *CDK2*, *YWHAE*, *CDC20*	3.47 × 10^−2^

*ABCA2*, ATP binding cassette subfamily A member 2; *ABCB9*, ATP binding cassette subfamily B member 9; *AIFM2*, Apoptosis inducing factor mitochondria associated 2; *AKT3*, AKT serine/threonine kinase 3; *ALDH18A1*, Aldehyde dehydrogenase 18 family member A1; *ALDH4A1*, Aldehyde dehydrogenase 4 family member A1; *ALDH7A1*, Aldehyde dehydrogenase 7 family member A1; *AMD1*, Adenosylmethionine decarboxylase 1; *ANGPT2*, Angiopoietin 2; *AP3B1*, Adaptor related protein complex 3 subunit beta 1; *ARRB1*, Arrestin beta 1; *ARRB2*, Arrestin beta 2; *ARSA*, Arylsulfatase A; *BAX*, BCL2 associated X, apoptosis regulator; *BBC3*, BCL2 binding component 3; *BCL2L1*, BCL2-like 1; *BIRC3*, Baculoviral IAP repeat containing 3; *BMS1*, BMS1 ribosome biogenesis factor; *BRCA1*, BRCA1 DNA repair associated; *CACNA2D1*, Calcium voltage-gated channel auxiliary subunit alpha2delta 1; *CARNS1*, Carnosine synthase 1; *CCNB1*, Cyclin B1; *CCND3*, Cyclin D3; *CD68*, CD68 molecule; *CD82*, CD82 molecule; *CDC20*, Cell division cycle 20; *CDC27*, Cell division cycle 27; *CDC6*, Cell division cycle 6; *CDK2*, Cyclin dependent kinase 2; *CDKN1A*, Cyclin dependent kinase inhibitor 1A; *CKB*, Creatine kinase B; *CSF1*, Colony stimulating factor 1; *CSF2RB*, Colony stimulating factor 2 receptor subunit beta; *CTSC*, Cathepsin C; *CTSD*, Cathepsin D; *CTSV*, Cathepsin V; *CTSZ*, Cathepsin Z; *CUL1*, Cullin 1; *CYBA*, Cytochrome b-245 alpha chain; *DBF4*, DBF4 zinc finger; *DDB2*, Damage specific DNA binding protein 2; *DDIT3*, DNA damage inducible transcript 3; *DDX39B*, DExD-box helicase 39B; *DDX5*, DEAD-box helicase 5; *DEGs*, Differentially expressed genes; *DFFB*, DNA fragmentation factor subunit beta; *DHX15*, DEAH-box helicase 15; *DKC1*, Dyskerin pseudouridine synthase 1; *DUSP1*, Dual specificity phosphatase 1; *DUSP5*, Dual specificity phosphatase 5; *E2F4*, E2F transcription factor 4; *EDN1*, Endothelin 1; *EPHA2*, EPH receptor A2; *FAS*, Fas cell surface death receptor; *FGFR3*, Fibroblast growth factor receptor 3; *FLNC*, Filamin C; *FOS*, Fos proto-oncogene, AP-1 transcription factor subunit; *FUCA1*, Alpha-L-fucosidase 1; *GADD45A*, Growth arrest and DNA damage inducible alpha; *GAMT*, Guanidinoacetate N-methyltransferase; *GAR1*, GAR1 ribonucleoprotein; *GBA*, Glucosidase, beta, acid; *GGT1*, Gamma-glutamyltransferase 1; *GNL3*, G protein nucleolar 3; *GPC1*, Glypican 1; *GSTM2*, Glutathione S-transferase mu 2; *GSTT2B*, Glutathione S-transferase theta 2B; *GUSB*, Glucuronidase beta; *HEATR1*, HEAT repeat containing 1; *HGF*, Hepatocyte growth factor; *HNRNPA1*, Heterogeneous nuclear ribonucleoprotein A1; *HNRNPA3*, Heterogeneous nuclear ribonucleoprotein A3; *HNRNPC*, Heterogeneous nuclear ribonucleoprotein C; *HNRNPK*, Heterogeneous nuclear ribonucleoprotein K; *HNRNPM*, Heterogeneous nuclear ribonucleoprotein M; *HNRNPU*, Heterogeneous nuclear ribonucleoprotein U; *HSP90AB1*, Heat shock protein 90 alpha family class B member 1; *HSPA8*, Heat shock protein family A (Hsp70) member 8; *HYAL1*, Hyaluronidase 1; *ICAM1*, Intercellular adhesion molecule 1; *IL1A*, Interleukin 1 alpha; *IMP4*, IMP U3 small nucleolar ribonucleoprotein 4; *ITPR2*, Inositol 1,4,5-trisphosphate receptor type 2; *ITPR3*, Inositol 1,4,5-trisphosphate receptor type 3; *JUN*, Jun proto-oncogene, AP-1 transcription factor subunit; *JUND*, JunD proto-oncogene, AP-1 transcription factor subunit; KEGG, Kyoto encyclopedia of genes and genomes; *KITLG*, KIT ligand; *KLF2*, KLF transcription factor 2; *LAMP3*, Lysosomal associated membrane protein 3; *LAPTM5*, Lysosomal protein transmembrane 5; *LGMN*, Legumain; *LMNB2*, Lamin B2; *LSG1*, Large 60S subunit nuclear export GTPase 1; *MAN2B1*, Mannosidase alpha class 2B member 1; *MAP3K4*, Mitogen-activated protein kinase kinase kinase 4; MAPK, Mitogen-activated protein kinase; *MAPK8IP1*, Mitogen-activated protein kinase 8 interacting protein 1; *MCM6*, Minichromosome maintenance complex component 6; *NAGA*, Alpha-N-acetylgalactosaminidase; *NAT10*, N-acetyltransferase 10; *NEU1*, Neuraminidase 1; *NF1*, Neurofibromin 1; *NFKB2*, Nuclear factor kappa B subunit 2; *NFKBIA*, NFKB inhibitor alpha; *NHP2*, NHP2 ribonucleoprotein; *NOP56*, NOP56 ribonucleoprotein; *NPC1*, NPC intracellular cholesterol transporter 1; *NPC2*, NPC intracellular cholesterol transporter 2; *NR4A1*, Nuclear receptor subfamily 4 group A member 1; *NRAS*, NRAS proto-oncogene, GTPase; *ORC6*, Origin recognition complex subunit 6; *P4HA2*, Prolyl 4-hydroxylase subunit alpha 2; *PARP3*, Poly(ADP-ribose) polymerase family member 3; *PCNA*, Proliferating cell nuclear antigen; *PDGFA*, Platelet derived growth factor subunit A; *PECAM1*, Platelet and endothelial cell adhesion molecule 1; *PGF*, Placental growth factor; *PIDD1*, P53-induced death domain protein 1; *PKMYT1*, Protein kinase, membrane associated tyrosine/threonine 1; *PLA2G4A*, Phospholipase A2 group IVA; *PLA2G4C*, Phospholipase A2 group IVC; *PLK1*, Polo-like kinase 1; *PMAIP1*, Phorbol-12-myristate-13-acetate-induced protein 1; *POLH*, DNA polymerase eta; *PPIL1*, Peptidylprolyl isomerase-like 1; *PPM1D*, Protein phosphatase, Mg2+/Mn2+ dependent 1D; *PRKACB*, Protein kinase cAMP-activated catalytic subunit beta; *PRPF19*, Pre-mRNA processing factor 19; *PRPF40A*, Pre-mRNA processing factor 40 homolog A; *PSAP*, Prosaposin; *PTK2*, Protein tyrosine kinase 2; *PTPN7*, Protein tyrosine phosphatase non-receptor type 7; *PTPRR*, Protein tyrosine phosphatase receptor type R; *PYCR1*, Pyrroline-5-carboxylate reductase 1; *RAC2*, Rac family small GTPase 2; *RAN*, RAN, member RAS oncogene family; *RASA1*, RAS p21 protein activator 1; *RELB*, RELB proto-oncogene, NF-kB subunit; *RHOC*, Ras homolog family member C; *RIPK1*, Receptor interacting serine/threonine kinase 1; *RPS6KA1*, Ribosomal protein S6 kinase A1; *RRAS*, RAS related; *RRM2*, Ribonucleotide reductase regulatory subunit M2; *SAT1*, Spermidine/spermine N1-acetyltransferase 1; *SDC1*, Syndecan 1; *SDC4*, Syndecan 4; *SESN1*, Sestrin 1; *SESN2*, Sestrin 2; *SMOX*, Spermine oxidase; *SMS*, Spermine synthase; *SNRPA1*, Small nuclear ribonucleoprotein polypeptide A’; *SNRPB*, Small nuclear ribonucleoprotein polypeptides B and B1; *SQSTM1*, Sequestosome 1; *SRM*, Spermidine synthase; *SRSF1*, Serine and arginine rich splicing factor 1; *SRSF10*, Serine and arginine rich splicing factor 10; *SRSF2*, Serine and arginine rich splicing factor 2; *SRSF3*, Serine and arginine rich splicing factor 3; *SRSF7*, Serine and arginine rich splicing factor 7; *STAG1*, Stromal antigen 1; *TAOK1*, TAO kinase 1; *TCERG1*, Transcription elongation regulator 1; *TGFB1*, Transforming growth factor beta 1; THBS1, Thrombospondin 1; *TOP2B*, DNA topoisomerase II beta; *TP53*, Tumor protein p53; *TP53I3*, Tumor protein p53 inducible protein 3; *TPP1*, Tripeptidyl peptidase 1; *TRA2A*, Transformer 2 alpha homolog; *TRA2B*, Transformer 2 beta homolog; *TRAF1*, TNF receptor associated factor 1; *TRPV4*, Transient receptor potential cation channel subfamily V member 4; *VCAM1*, Vascular cell adhesion molecule 1; *WDR3*, WD repeat domain 3; *WDR43*, WD repeat domain 43; *XPO1*, Exportin 1; *XRN2*, 5′-3′ exoribonuclease 2; *YWHAE*, Tyrosine 3-monooxygenase/tryptophan 5-monooxygenase activation protein epsilon.

**Table 3 nutrients-14-04518-t003:** Genes superimposed by multiple datasets.

Gene Symbol	Log2 (FC)	*q*-Value	Function	Refs.
*LRP1*	1.757	8.32 × 10^−6^	(+/−) Atherosclerosis (−) CCR7 (+) PPARγ	[40,41,42]
*CCDC92*	0.813	8.32 × 10^−6^	(+) insulin resistance	[32,43]
*TRIB1*	0.679	4.36 × 10^−2^	(+) OLR1 (+) oxLDL uptake (+) the formation of lipid-laden foam cells	[44]
*TGFB1*	0.556	2.08 × 10^−2^	(+) Atherosclerosis	[45]
*PECAM1*	0.403	4.88 × 10^−2^	(−) blood T-cell activation	[46]
*SERPINH1*	−0.387	2.90 × 10^−2^	relate to Atherosclerosis	[47,48,49]
*PHACTR1*	−0.950	2.01 × 10^−2^	(+) EDN1	[14,50]

*CCDC92*, Coiled-coil domain containing 92; CCR7, C-C motif chemokine receptor 7; EDN1, Endothelin 1; *LRP1*, LDL receptor related protein 1; OLR1, Oxidized low density lipoprotein receptor 1; oxLDL, Oxidized low-density lipoprotein; *PECAM1*, Platelet and endothelial cell adhesion molecule 1; *PHACTR1*, Phosphatase and actin regulator 1; PPARγ, Peroxisome proliferator activated receptor gamma; *SERPINH1*, Serpin family h member 1; *TGFB1*, Transforming growth factor beta 1; *TRIB1*, Tribbles pseudokinase 1.

## Data Availability

GEO repository with the accession number: GSE211883.

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
