# Peer review of "Genome-Wide Transcriptional Profiling Reveals PHACTR1 as a Novel Molecular Target of Resveratrol in Endothelial Homeostasis"

_nutrients, 2022, doi:10.3390/nu14214518_

Round 1

Reviewer 1 Report

The manuscript of Meiming Sui et al. entitled: “Genome-wide transcriptional profiling reveals PHACTR1 as a 2 novel molecular target of resveratrol in endothelial homeostasis” attempted to define the role of resveratrol in endothelial homeostasis.

My comments are listed below:

1)Methods section provides only some general background and lacks basic information, such as methods for RNA sequencing (RNA-seq). Instead of this reserch [20 ] is cited  S. Xu, Y. Liu, Y. Ding, S. Luo, X. Zheng, X. Wu, Z. Liu, I. Ilyas, S. Chen, S. Han, P. J. Little, M. K. Jain, J. Weng, The zinc finger 444 transcription factor, KLF2, protects against COVID-19 associated endothelial dysfunction. Signal Transduction and Targeted Ther-445 apy 2021, 6.

2) the quality of the figures is insufficient

3) Authors  treated endothelial cells with different 270 doses of RES for 12 h, followed by adenovirus infection for another 24 h to overexpress 271 PHACTR1. (only some general background and lacks basic information, such as methods ) The purpose of examining should be presented more clearly to the reader. Its legitimacy was not clarified, it was not referred/discussed in the discussion section.

4) Authors found that RES was able (line 226) to upregulate the mRNA level of KLF2 and downregulate the mRNA level of EDN1 in a concentration-dependent manner. Why are these results not confirmed by real-time, Western Blotting analysis Are they less important than the effect on PHACTR1? Why?

Reviewer 2 Report

The manuscript investigated the possible role of PHACTR1 and resveratrol in endothelial homeostasis. Overall, it is an interesting study, but need some clarifications.

1.      There is no need to deal with well-known facts in the Introduction. Instead, add more details regarding the PHACTR1. E.g. what is its physiological role?

2.      Please explain each of your abbreviations the first time it appears in the main text. Do not define abbreviations that are not used. In addition, too many abbreviations are used in this manuscript.

3.      What does it mean - shear stress mimetic drug?

4.      Add all obligatory data in the legends to tables and figures - they are absolutely insufficient.

5.      Line 45 – why have you used capitals in Oxide Synthane?

6.      Lines 76-78 – better fit into discussion

7.      Line 81 – what does it mean „normal“? Please, be preciese

8.      Line 90 - the concentration of RES should be specified here

9.      Line 91 – for ,,some“ experiments? Please, be more specific

10.  Line 133 - it could be specified what proteins were analyzed

11.  Line 145 - how was the data normality checked?

12.  Lines 153-161 – not results, better for methodology section

13.  Lines 175 - 180 - not results, better for Discussion section

14.  Figure 1 A – I am no table to read the text, it is too small

15.  Figure 1 B – I am missing p-values for the r-values

16.  Table 1 – row Log2(FC) – are 9 decimals needed?

17.  Figure 2C – no chance to read – too small text

18.  Figure 2D – text could be also better readable

19.  In the discussion, the confrontation of obtained data with those available from human studies is missing.

Round 2

Reviewer 2 Report

In statistical analysis, the mention regarding the usage of one-way ANOVA (+ post hoc test) is missing.

Too many abbreviations are used in this manuscript and not all of them are explained in the figure/table captions.

I would also like to see the authors discuss their findings more with those available from larger human studies (in which resveratrol was administered). It is important for understanding the importance of obtained data, and their possible clinical significance.

According to my opinion, it is important to logically organize the text in the manuscript in the main chapters.
